# Mechanical Strength of Additive Manufactured and Standard Polymeric Components Joined Through Structural Adhesives

**DOI:** 10.3390/polym16213036

**Published:** 2024-10-29

**Authors:** Andrea Spaggiari, Simone Orlandini

**Affiliations:** Department of Science and Methods for Engineering, University of Modena and Reggio Emilia, Via G. Amendola 2-Pad. Buccola, 42122 Reggio Emilia, Italy; 285816@studenti.unimore.it

**Keywords:** polymeric additive manufacturing, bonded joints, pin–collar tests, structural adhesives

## Abstract

The main aim of this work is to evaluate the mechanical properties of additive manufactured polymeric parts joined with standard plastic parts through structural adhesives. The primary advantage of this technique is its ability to significantly increase the size of the final assembly by using additive manufacturing (AM) for complex joints and inexpensive, reliable extruded plastic parts for load-bearing components. This hybrid assembly combines the flexibility and shape adaptability of AM with the structural strength and cost-effectiveness of extruded polymer parts, resulting in a final design that performs comparably to the base material. The materials used in the paper are rigid acrylic adhesive and toughened acrylic, both applicable with almost no surface preparation and fast curing. The 3D-printed parts are produced in ABS, while the standard parts are in PVC. First, the work is devoted to estimating the performance of the adhesives using pin–collar joints and a combined numerical and experimental methodology. The second section presents and discusses the results of two more realistic applications of adhesive bonding to hybrid complex joints. For the pin–collar joints, the results show failure mostly in the adhesive, with an average shear stress of 11.5 MPa and 5.22 MPa and a stiffness of 4449 N/mm and 3649 N/mm for the rigid and toughened adhesives, respectively. The results of the adhesive bonding of structural joints show that the adhesive is always capable of providing the load-carrying capacity required to achieve the strength of traditionally manufactured polymeric parts. The paper shows that adhesives are a feasible way to expand the potential of 3D-printed equipment to obtain larger hybrid parts partially realized with traditional technology, especially with inexpensive off-the-shelf bars and sections.

## 1. Introduction

The growing utilization of lightweight materials like carbon or glass fiber-reinforced composites has promoted the adoption of adhesive bonding technology as a reliable approach for assembling various components in mechanical systems, especially in industries like aerospace and automotive, where stiffness and weight are critical. Adhesives offer several advantages compared to traditional mechanical joining techniques such as welding, riveting, or threaded connections. Firstly, they eliminate the need for holes in the substrate, which could compromise the structural integrity of the fibers. Secondly, they enable the designer to join different materials while ensuring a fairly uniform distribution of loads along the entire bondline. Lastly, adhesive bonding is increasingly compatible with automation, a crucial factor in reducing manufacturing costs [1,2]. Conversely, some weak points can be ascribed to this methodology, and the most important one is the almost mandatory need for surface preparation [3,4,5,6], which is always undesirable for industrial players since it involves contamination problems and longer processing times. Moreover, the typical strong elastic modulus difference between the adhesive and adherends causes strong stress peaks at the adhesive layer borders, promoting premature failure of the joint. In the technical literature, alternatives such as special test specimens with low-stress concentrations can be found: the napkin ring [7], the Iosipescu specimen [8,9], and a four-point bending test [10,11], which provide a shear stress state in the adhesive layer with no or moderate stress concentrations. The situation is even worse considering the peel direction, where the strong difference in the elastic properties of the materials causes more severe stress concentrations when the adhesive is deployed in thin layers [12,13,14]. The scientific literature reports some methods to lower the degree of the singularity of these peaks, such as a slight modification of the adherends [15,16,17], or spew fillet [18,19], the Arcan test [20,21,22], and relief grooves [23,24].

By incorporating contemporary methods and considerations for additive manufacturing (AM), certain authors have explored the potential of diminishing stress concentrations. This asset can be achieved by either decreasing the stiffness of the adherends or enhancing the adhesive stiffness using functionally graded materials [25,26,27,28,29], but the industrial application of these techniques is limited since the technology is not yet ready to implement these findings in a productive context. Despite the limitations, AM has gained widespread acceptance and could prove beneficial in addressing certain challenges. While techniques like selective laser sintering, selective laser melting, or electron beam are widely used for the AM of metal components, the prevalent method for polymers remains fused filament fabrication. This approach is cost-effective, relatively quick, and results in finished 3D-printed parts with a satisfactory surface finish. Components produced through AM offer the advantage of extreme design flexibility, making this technique ideal for enhancing “by design” the performance of adhesive-bonded joints in terms of strength compared to traditional solutions. In this context, the stiffness of metal AM parts can be intentionally reduced without compromising external geometry. This feature can be achieved by incorporating metamaterial concepts such as lattice structures or hollow components. This approach could result in a comparable stiffness level between the adhesive and the adherends, contributing positively to lightweight design. Furthermore, the combination of additive manufacturing (AM) and adhesive bonding offers a solution to two of the current limitations of AM technology: the restricted working volume of AM machines and the specific cost per part. On the one hand, many 3D printers excel at producing small-scale components, but with an increase in volume, material distortions emerge, the stability of thin-walled structures is not easily obtained, and the precision decreases. On the other hand, the cost per part for polymeric additive manufacturing is around $20–40 USD [30] since 1 kg of ABS filament material is around $20–25, while the same amount of ABS ranges from around $2–3 per kg if standard bars or sheets are considered according to commercial suppliers (https://www.plastock.co.uk/, accessed on 24 October 2024). The increase of 10 times in price motivates the use of AM only where strictly needed and creates the research motivation for a proper joining technique between solid polymeric parts obtained by traditional manufacturing (i.e., extruded bars) and complex AM parts. Structural adhesive could play an important role in this hybrid junction, extracting the maximum values from both technologies. In the literature, some work has explored the use of metal–polymer or metal–carbon fiber joints [31,32]. These joints are indeed interesting, but they present the same problem as the traditional bonded joint, which is the high elastic mismatch between the adherend and the adhesive that creates severe stress concentrations at the edges, undermining the joint performance. Several studies [33,34,35,36,37,38,39,40] have already investigated the behavior of the adhesive bonding between AM parts, demonstrating that the mechanical resistance of the adhesively bonded AM joint could reach the same level as the base polymeric material, thus obtaining the best mechanical performance possible for an AM part. Starting from this solid foundation, this paper aims to analyze the performance of adhesive in bonding traditionally manufactured polymeric bars and AM parts, keeping the benefit of a low elastic mismatch between the adherend and adhesive and increasing the performance of the non-3D-printed adherend. The first part of this paper is devoted to the assessment of the critical stress of the adhesive applied to the pin–collar joints [41,42] ASTM D4562-01, ISO-10123) already used for press-fitted and adhesive joints of metallic parts [43,44], which is particularly effective for this purpose since it limits the problem of the substrate failure already found in other research [34]. The second part of the paper applies the adhesive to a couple of possible hybrid joints to estimate the load-carrying capacity in two coupons, which resembles a more realistic application. This paper aims to evaluate the pin–collar joints as a fast and feasible way to characterize the bonding performance of acrylic adhesive with traditionally manufactured parts and AM parts, as this information cannot be traced in the technical literature, and the present work aims to increase scientific knowledge in this area.

## 2. Materials and Method

### 2.1. Pin–Collar Joints

#### 2.1.1. Experimental Test Procedure

The pin–collar specimens were designed as follows: Two different materials obtained from two different technologies were bonded. The pin was obtained by cutting a 10 mm cylindrical bar of PVC, which was experimentally characterized before the specimen’s preparation. The results of the tensile test on the bar provided an elastic modulus of the PVC of 790 MPa (±4.2%) and a maximum tensile strength of 41.8 MPa (±1.8%). The data for the PVC are reported in Appendix A. These values are lower than expected, as many PVC objects are more rigid. Regardless, for the sake of the research, the elastic modulus of the parts is not fundamental since it was verified experimentally. The collar was produced by AM using a Stratasys Fortus 250 MC printer [45], which grants a reliable repeatability of the specimens, considering a minimum layer precision of 0.254 mm with the collar axis in the Z direction. All the specimens were printed at quasi-full density in order to achieve the best mechanical properties of the polymer (around 33 MPa according to the producer [46]). A set of additional dogbone specimens was printed together with the adherends and tested to verify this value. The data for the ABS dogbone specimens are reported in Appendix A. An elastic modulus of 991.5 MPa and a maximum tensile stress of 35.9 MPa were retrieved, which is in good agreement with the datasheet indications. The internal diameter of the pin is the nominal diameter of the bar, the external diameter of the collar is 21 mm, and the collar height is 9 mm. The specimen geometry is adapted based on the ASTM standard by proper scaling of each dimension. Two acrylic adhesives were chosen to bond the specimens. The first one is Loctite 401 [47], a rigid cyanoacrylic adhesive already tested in previous studies that provides, according to the technical datasheet, a lap shear strength on ABS above 7.5 MPa and on PVC above 10 MPa. The second adhesive is Loctite 480 [48], a toughened cyanoacrylic adhesive that provides, according to the technical datasheet, a lap shear strength on ABS above 6 MPa and on PVC above 4 MPa. The elastic modulus of the adhesive is not provided by the producer, but according to the Materials Handbook [49], their Young modulus is in the 1000–1400 MPa range. According to the product formulation, the Loctite 480 is more compliant, while the Loctite 401 is stiffer; therefore, we considered two different elastic moduli, E_480_ = 1000 MPa and E_401_ = 1400 MPa. The Poisson’s ratio was 0.3 for all the parts.

On the experimental side, five replicates were carried out for each configuration, which grants good statistical support according to the Design of Experiment theory [50]. All the specimens were carefully measured before bonding to estimate the clearance between the pins and the collars. It is important to highlight that a certain tolerance is present both in bar roundness and in AM parts; therefore, the coupling cannot have constant clearance. Moreover, the AM collars were not sandblasted nor gritblasted before the bonding since the aim is to reduce the amount of surface preparation and to exploit the AM parts as produced by the printer, while the pins were lightly prepared using sandpaper, as suggested in [51]. Then, dust and impurities were removed with a clean cloth using the Loctite Cleaner 7063 on both parts. The test rig used to bond the specimens and a picture of the coupons after bonding are shown in Figure 1a,b, and the specimens under compression testing are shown in Figure 1c. All the specimens were cured at room temperature and relative humidity of 50% for 24 h, which largely exceeds the prescribed polymerization time of 5 min on ABS and PVC [47]. Figure 1c shows the experimental set-up where a quasi-static displacement was applied to the specimens using a universal tensile machine (Galdabini Quasar 25), equipped with a 25 kN load cell. The applied crosshead displacement is 1 mm/min to avoid possible viscoelastic effects, both for the adherends and the adhesive.

#### 2.1.2. Finite Element Models of Pin–Collar Joints

The numerical finite element models were designed using Abaqus (DS Simulia Established Products 2023) and exploiting the axisymmetric shape of the system, as shown in Figure 2. The model was intended to mainly provide a comparison between the performance of the two adhesives; therefore, all the materials were modeled as perfectly elastic with the above-described properties. The mesh was refined in the adhesive corners by placing six elements in the adhesive thickness (Figure 2—right detailed view), but the well-known singularity at the bondline corners prevents a quantitative assessment of their properties by simply assessing the stress using FEA.

The mesh details are the following: 192,828 nodes, 189,540 elements, average dimension in the bondline: 0.01 mm, linear axisymmetric plane elements, and reduced integration (CAX4R). The model is supported below the collar in the vertical direction, and the load is applied through a reference point (RP-1) in the top part of the pin, as shown in Figure 2. In order to find the critical stress for the adhesive, the applied load is based on the average experimental load, as described in the results and discussion section.

### 2.2. Hybrid AM–Traditional Bonded Joints

Once the static strength of the ABS parts in AM bonded with PVC is retrieved through experimental tests, two different joints that resemble more practical applications are numerically modeled and experimentally tested, one based on cylindrical PVC bars and the other on square PVC sections.

#### 2.2.1. Cylindrical Double Hub Joint

The first joint exploits the same cylindrical bar used in the pin–collar test, while the AM joint is shaped as a double hub for the bar. The schematic of the Cylindrical Double Hub Joint (CDHJ) is reported in Figure 3a, while Figure 3b reports the AM parts after printing. Two replicates of the joints were manufactured, and the parts were bonded only with the Loctite 401 adhesive since it was the one with better performance in the pin and collar joints. The lower amount of experimental test compared to the 5 replicates of the pin-collar ones is motivated by the fact that these joints are not exploited to retrieve the adhesive properties, but only to validate them on joint which could be more similar to a real application of this approach.

These kinds of joints can be tested both in tension and torsion, but according to the adhesive mechanics, tensile tests are more challenging for the adhesive. The torsion test, as already reported by many researchers [1,25,52,53], led to shear stress in the adhesive with almost no stress concentration. The tensile tests, on the other hand, are insidious since, due to Poisson’s effect, there is not only shear stress on the lateral bonded face but also peel stresses, as thoroughly reported in the literature [52,53]. Therefore, to fulfill the aim of pursuing a more realistic and challenging loading condition, we applied a tensile force as shown in Figure 3c.

Moreover, a finite element model of the CDHJ was developed with Abaqus, and again the model exploited the axial symmetry to decrease the computational effort. In addition, the model is symmetric with respect to the horizontal centerline, so only one-half of the axisymmetric joint is modeled. The mesh dimension is comparable to the previous model, with six elements in the adhesive bondline. The mesh details are reported in Figure 4, and the applied load is the average experimental maximum load in accordance with the pin–collar joints procedure.

#### 2.2.2. Prismatic Double Hub Joint

The second type of realistic joint analyzed resembles the first one, but instead of a cylindrical bar, a prismatic PVC bar with a 20 mm side square section is used. The schematic of the Prismatic Double Hub Joint (PDHJ) is reported in Figure 5a, while Figure 5b reports the AM parts after printing. The PDHJ was manufactured, and the parts were bonded again with Loctite 401 adhesive to be consistent with the tests described in Section 2.2.1. Similarly to the CDHJ, this joint can also sustain load in many directions, including torsional and bending moments, in addition to the axial load. The first two loading conditions are frequently encountered in practical applications, and the PDHJ is able to provide good structural strength due to its prismatic shape. While the adhesive does not play a crucial role, it is exploited to keep the system tight and joined. Therefore, in order to provide the most stressful condition in the bonded region, we also tested the PDHJ using a tensile set-up as with the CDHJ, as shown in Figure 5c, in order to stress both the adherends and the adhesive.

Moreover, for this configuration, a finite element model was also developed with Abaqus. In this case, it was not possible to exploit the axial symmetry, but the three planes of symmetry were used to decrease the computational effort, which is much higher since this is an intrinsic 3D model. The mesh dimension is comparable to the previous models described in Section 2.1.2, with six elements in the adhesive bondline. The mesh details are reported in Figure 6, and the applied load is once again the average experimental maximum load in accordance with the pin–collar joints procedure.

## 3. Results and Discussion

### 3.1. Pin–Collar Joints: Experimental Tests

The experimental curves retrieved for the pin–collar joints are reported in Figure 7. The blue curves show the behavior of the Loctite 401 joints, while the red curves show the behavior of the Loctite 480 joints. There is a clear difference between the first and the second adhesive, especially in terms of maximum force developed by the joint, while the stiffness of the system is comparable. The failure modes are reported in Figure 8, where it is evident that the best-performing adhesive (Loctite 401) could lead to substrate failure, while the Loctite 480 always fails in adhesive cohesive mode.

By simply dividing the maximum experimental force by the lateral bonded area, it is possible to retrieve the average shear stress while calculating the slope of the curve in the first linear part of the chart in Figure 7, which will give the stiffness of the joint. This procedure leads to the results in Table 1.

### 3.2. Pin–Collar Joints: Finite Element Analyses

The finite element model described in Section 2 is exploited only to compare the two adhesives in terms of structural stresses. The von Mises stress map of the pin–collar joint is shown in Figure 9a for the Loctite 401 and in Figure 9c for the Loctite 480, while the detail of the most stressed corner (in terms of maximum principal stress) is reported with magnification in Figure 9b–d for Loctite 401 and 480, respectively. The equivalent stress contours provide a global overview of the stress state, while the maximum principal stress is used to extract the so-called “*structural stresses*” [54]. The structural stresses can be used to estimate the joint performance, especially in a comparative way since they can be used for the construction of a failure envelope of the joint considering both the peel and shear components. The “*structural stress*” is measured in the middle of the bondline thickness in order to avoid the corner singularity. The peak structural stresses obtained in the bondline thickness for these adhesives are based on the maximum principal stress, which is more reliable than the von Mises stress since adhesives, unlike metals, are sensitive to hydrostatic components of the stress tensor and, especially for cyanoacrylates, tend to assume a brittle behavior. To evaluate the correctness of the numerical model, the maximum vertical displacement of the pin was computed for both adhesives, and the results are reported in Table 2. By comparing the results of Table 1 and Table 2, it is evident that the simple FE model adopted can convey the mechanical behavior of the pin–collar joints, providing a fair estimate of the displacement at failure, with a low error of −0.5% for Loctite 401 and below 10% for Loctite 480.

### 3.3. Cylindrical Double Hub Joint: Experimental Test

The experimental curves retrieved for the CDHJ are reported in Figure 10. Both curves refer to joints bonded with Loctite 401, which was the best-performing adhesive. The peak average load of the CDHJ is 2129 N. This average load can be used to compute the average stress both in the net tubular section of the hub and on the lateral bonded area of the adhesive according to the following formulae:$$\sigma_{av_{adherend}}=\frac{4F_{av}}{\pi \left(D^{2}-d^{2}\right)}=\frac{4\times 2129}{\pi \left(21^{2}-10^{2}\right)}=7.95\;MPa$$
(1)$$\sigma_{av_{adhesive}}=\frac{F_{av}}{\pi dL}=\frac{2129}{\pi \times 10\times 15}=4.5\;MPa$$

As expected, the average stresses cannot be used to extract information on the joint behavior. It is useful therefore to analyze the tendency of the curves and the fracture surfaces. The two curves present a similar trend, with a linear elastic behavior of the system in the initial part and a non-linear behavior in the second part. The blue curve test ended due to additive manufactured part failure in the middle of the hub, while the orange curve ended due to failure near the end of the PVC pin bondline. A picture of the two failure modes is reported in Figure 11. In both cases, the adhesive was able to transfer the load between the pin and the AM parts. The position of the AM part failure was near the change in the section of the hub, where there is a stress concentration that possibly triggered the crack in the AM part.

### 3.4. Cylindrical Double Hub Joint: Finite Element Analyses

The FE analysis of the CDHJ, reported in Figure 12, can be used to clarify the experimental test results. The global stress state in the joint is quite complex, as shown in Figure 12a (von Mises contour), with stress concentration arising at both ends of the bondline. In Figure 12b, the principal structural stress is reported with a focus on the bondline upper edge, which is the most stressed area for the adhesive. By comparing the structural stress (i.e., picking the same node in the middle of the adhesive bondline), it is possible to compare the structural stress in the CDHJ (51.2 MPa) with the same value reported in Table 2 (81.33 MPa). Thus, the stress in the adhesive bondline is high but not high enough to trigger adhesive failure. On the contrary, by considering the von Mises stress in the AM lower edge, as shown in Figure 12c, the gray area exceeds the yield stress of the ABS substrate (33 MPa). Therefore, the failure of the AM hub starting from the lower corner, as shown in Figure 11, is confirmed by the stress analysis of the joint. It is now possible to compare the experimental test results and the numerical ones for the CDHJ as shown in Table 3, where it can be seen that the displacement at failure of the numerical FE model is consistent with the experimental data with an error of around 15% (3.6 mm vs. 3.1 mm). The applied load equals the experimental one, but this discrepancy was expected since the FE model is linear elastic, while deformation in the nonlinear regime happens in the experiments, as shown by the experimental curves in Figure 10.

Table 3 also reports the factor of safety of the components, expressed as the ratio between the peak stress in the adherend and the adhesive divided by the critical stress, which is the structural stress for the adhesive and the yield stress for the adherends. The result shows a factor of safety below 1 for the AM parts and above 1 for the adhesive, confirming the experimental results. The most relevant aspect of this approach is related to the fact that the simple model adopted, based on the structural stresses, can estimate the failure point (which occurs in the AM parts) by a simple comparison between the critical stress of the adherends and adhesives.

### 3.5. Prismatic Double Hub Joint: Experimental Test

The experimental curve retrieved for the PDHJ is reported in Figure 13. As for the CDHJ, the joint was bonded with Loctite 401 only. The curve shows a nearly linear behavior, and the image of the fracture surface highlights that the joints failed again due to poor adherend performance rather than the adhesive bonding. This behavior is surprisingly good for a general-purpose adhesive that carries the load between the AM parts and traditionally manufactured parts in a very efficient way. The maximum load of 5333 N corresponds to an average tensile axial stress in the PVC square bar of 13.55 MPa, much lower than its yield stress (41.8 MPa), which supports the fact that, in this case, the crack seems to start from the AM parts in ABS, triggered again by the stress concentration of the bonded joints. The most important difference between the CDHJ and the PDHJ is the position of the failure crack. In the CDHJ, the crack started at the bottom of the joint, while in this case, the crack started at the top. In both cases, the AM failed with an average load on its net section that was lower than its yield stress (around 10 MPa in this case). However, the stress concentration due to the bonding is enough to trigger the first crack and the subsequent joint failure, as shown by FEA.

### 3.6. Prismatic Double Hub Joint: Finite Element Analyses

The FE analysis of the PDHJ, reported in Figure 12 and Figure 14, reports the global stress state (Figure 14a, von Mises contour) with stress concentration arising in both ends of the bondline. In Figure 14b, the principal structural stress is reported with a focus on the bondline upper edge, which is the most stressed area, both for the adherend and adhesive. By following the same procedure as the CDHJ, we can compare the structural stress (41.52 MPa) with the same value reported in Table 2 (81.33 MPa). Thus, the stress in the adhesive bondline is high but not high enough to trigger adhesive failure. On the contrary, the von Mises stress in the PVC corner is above 50 MPa, and the yield stress of the PVC substrate is 41.8 MPa. Therefore, the failure of the AM hub starting from the upper corner, as shown in Figure 13, is confirmed by the FE analysis of the joint. Regarding the prediction of the displacement, the finite element model provides a 3.23 mm axial displacement, which is in fair agreement with the experimental data of 3.69 mm, given the simplicity of the linear elastic model adopted. This final test aims to verify that the adhesive is a reliable way to bond AM polymeric parts and traditional polymeric bars without lowering their performances in terms of load-carrying capacity. To this end, this final experiment on the PDHJ showed that the combination of AM parts and adhesive can lead to good structural efficiency with savings in costs and weight of the assembly without compromising stiffness or strength.

## 4. Conclusions

The current study presents findings on the integration of additive manufacturing polymer technology with traditional extruded polymer bars using structural adhesives. The initial phase of the research aims to identify a straightforward mechanical test to assess and estimate critical stress for the adhesive. The pin-collar test, typically employed for anaerobic adhesives, proves to be an effective method for this evaluation. To estimate the adhesive’s critical stress, a literature-based criterion focusing on “structural stresses” is utilized, which shows promise in characterizing the bonding and predicting joint performance. An axisymmetric finite element model is employed to derive critical structural stress values from preliminary tests on pin-collar joints using two different cyanoacrylate adhesives. The adhesive’s performance on pin-collar specimens highlights a very good behaviour of Loctite 401 adhesive compared with the Loctite 480 providing both higher peak force and displacement at failure. Another important feature is that the pin-collar specimen mostly causes failure in the adhesive rather than in the AM parts. Therefore, two more practical joint designs—a cylindrical and a prismatic double-hub joint—are fabricated and bonded only with Loctite 401. Test results from the cylindrical double-hub joint indicate that failure occurs within the additive manufacturing (AM) components rather than in the adhesive itself. Further finite element analysis reveals that the adhesive experiences lower stress under experimental conditions compared to the critical structural stress derived from pin-collar joints, while the stresses in the AM components exceed the yield strength of the material due to stress concentrations at the bond line. This same pattern is observed in the prismatic double-hub joint, where failure arises from cracks in both the AM part and the PVC bar. The findings suggest that adhesive bonding, along with a simple stress-based criterion identified in technical literature, could effectively guide the design of AM-bonded joints. Additionally, the performance of these hybrid bonded joints is found to be comparable to that of the base materials, with no adverse effects on joint stiffness or strength due to the adhesive. Another noteworthy finding is that using pin-collar joints can prevent failures in the AM substrate, thereby providing a more reliable design value for the critical shear strength of the adhesive used. The combination of AM components with traditional extruded bars has significant potential to enhance the capabilities of AM polymer technologies for large assemblies without necessitating expensive equipment, leveraging AM for complex joints while relying on conventional methods for simpler parts.

## Figures and Tables

**Figure 1 polymers-16-03036-f001:**
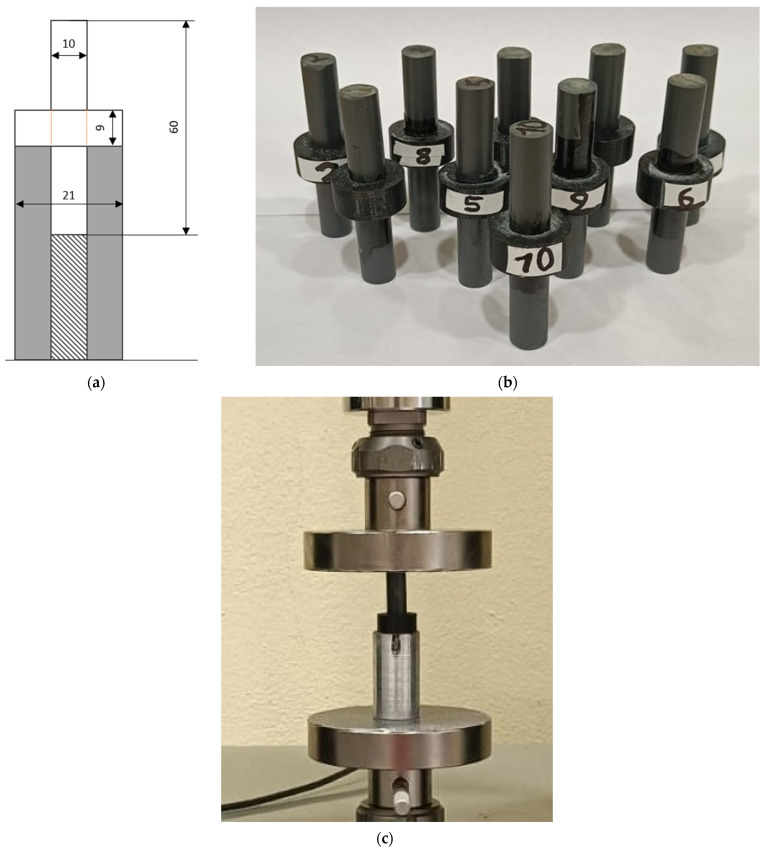
Main dimensions (in mm) of the pin–collar joint in a schematic drawing with the bondline in orange (**a**), a photo of the complete set of specimens after curing (**b**), and an experimental compression test on the pin–collar joint (**c**).

**Figure 2 polymers-16-03036-f002:**
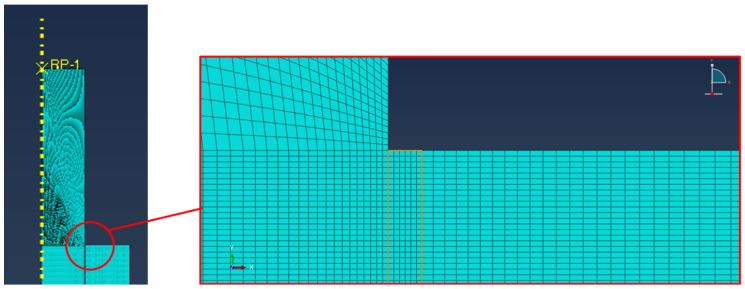
Detail of the FE mode of the pin–collar joints with a magnification of the adhesive bondline.

**Figure 3 polymers-16-03036-f003:**
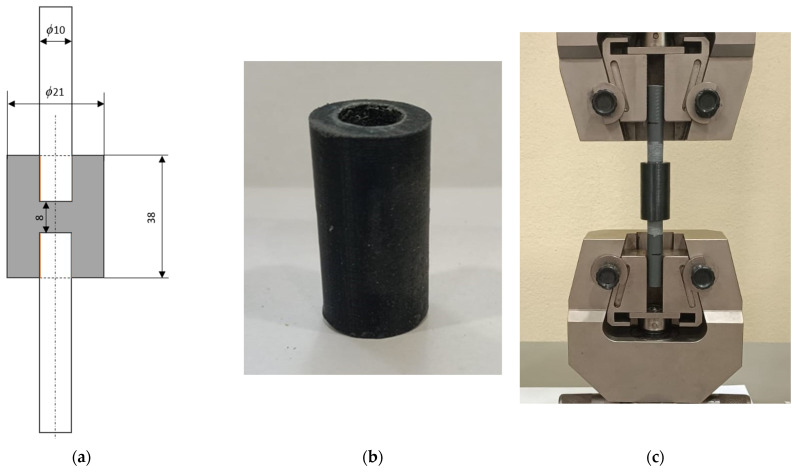
Schematic section of the CDHJ with dimensions in mm, not to scale (**a**), AM hub after printing (**b**), and tensile test set-up of the structure (**c**).

**Figure 4 polymers-16-03036-f004:**
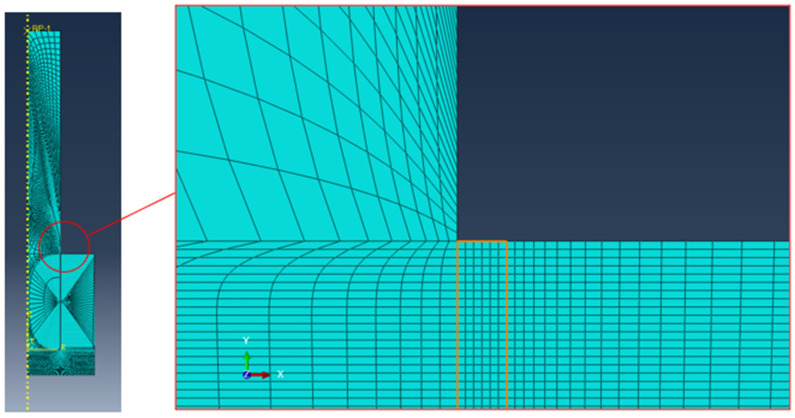
Detail of the FE model of the CDHJ, on the **left**, with a magnification view (on the **right**) of the adhesive bondline (highlighted in orange).

**Figure 5 polymers-16-03036-f005:**
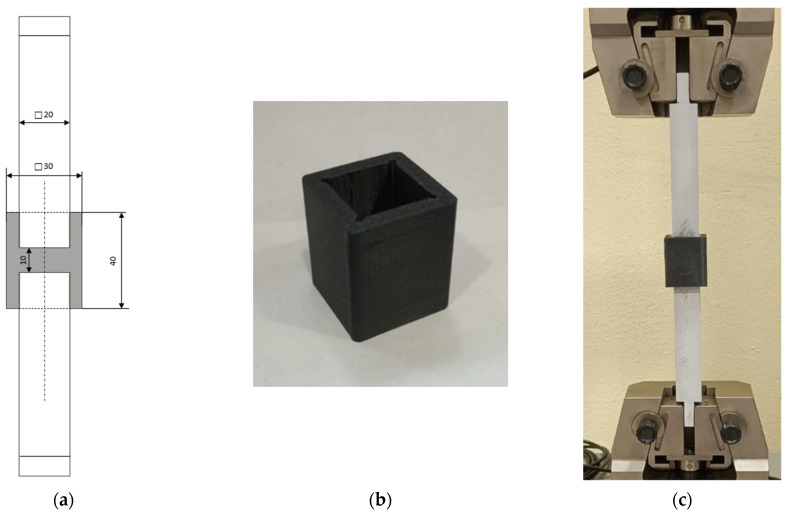
Schematic section of the CDHJ with dimensions in mm, not to scale (**a**), AM prismatic hub after printing (**b**), and tensile test set-up of the structure (**c**).

**Figure 6 polymers-16-03036-f006:**
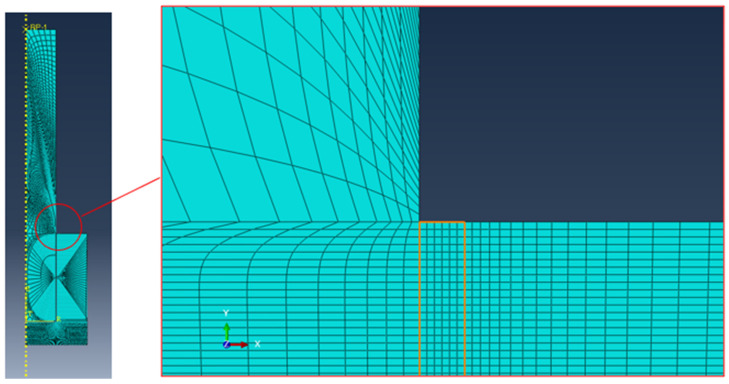
Detail of the FE model of the PDHJ with a magnification of the adhesive bondline (highlighted in orange).

**Figure 7 polymers-16-03036-f007:**
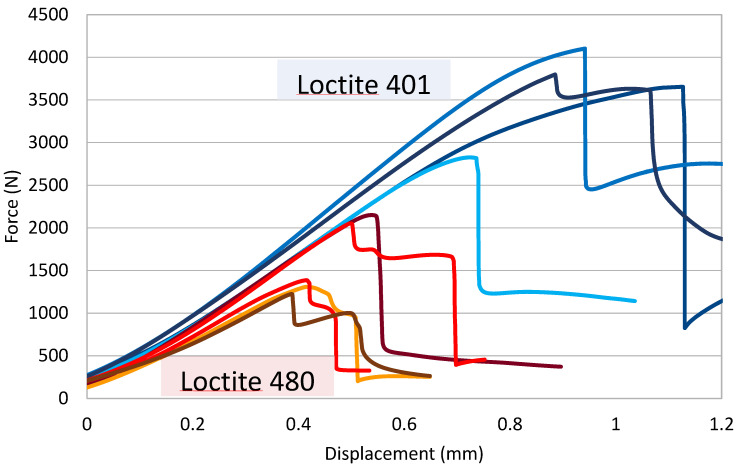
Force—Displacement curves of the pin-collar joints bonded with Loctite 401 (bluish curves) and with Loctite 480 (reddish curves).

**Figure 8 polymers-16-03036-f008:**
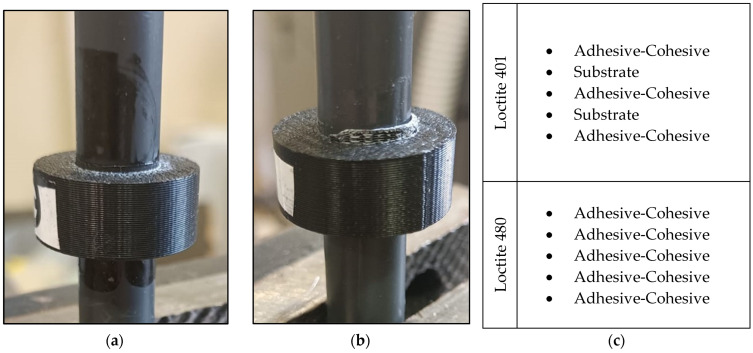
Adhesive cohesive failure mode (**a**), substrate failure mode (**b**) and table with experimental failure mode results (**c**) for the five replicates bonded with Loctite 401 and with Loctite 480.

**Figure 9 polymers-16-03036-f009:**
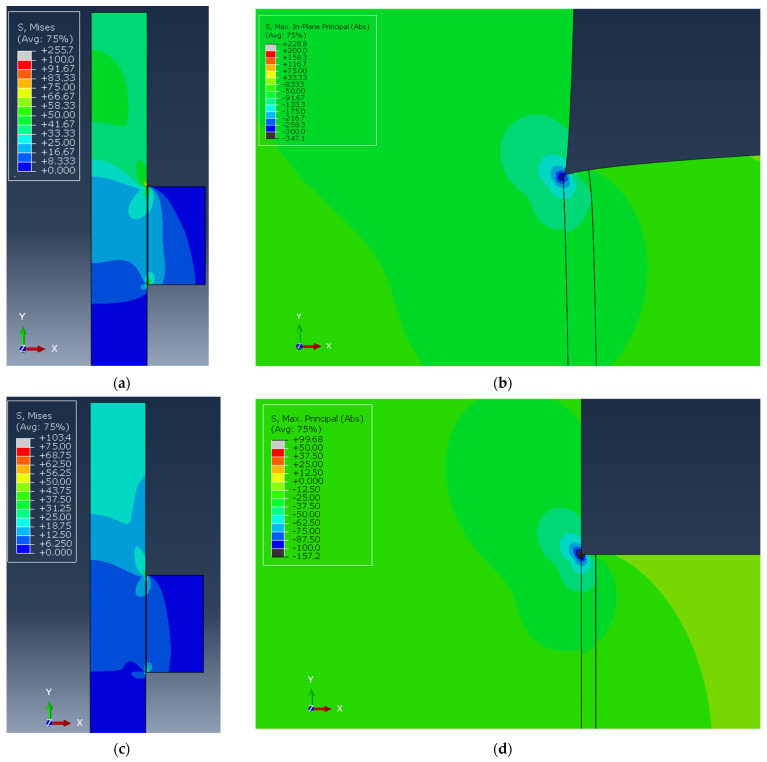
Von Mises stress maps of the pin–collar joints for Loctite 401 and 480 (**a**,**c**) and absolute maximum principal stress map of Loctite 401 and 480 (**b**,**d**) in the adhesive upper corner.

**Figure 10 polymers-16-03036-f010:**
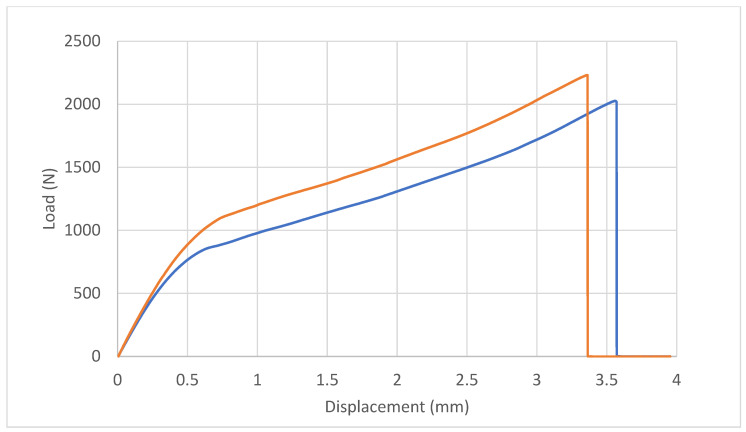
Experimental force–displacement curves of the two replicated of cylindrical double hub joint bonded with Loctite 401.

**Figure 11 polymers-16-03036-f011:**
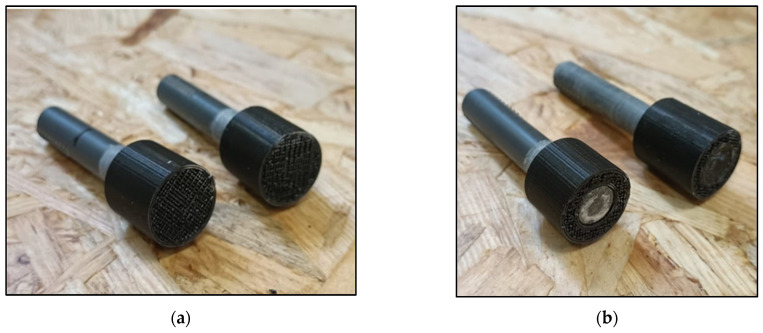
Failure modes of the CDHJ: AM part failure (**a**) and failure near the PVC pin end (**b**).

**Figure 12 polymers-16-03036-f012:**
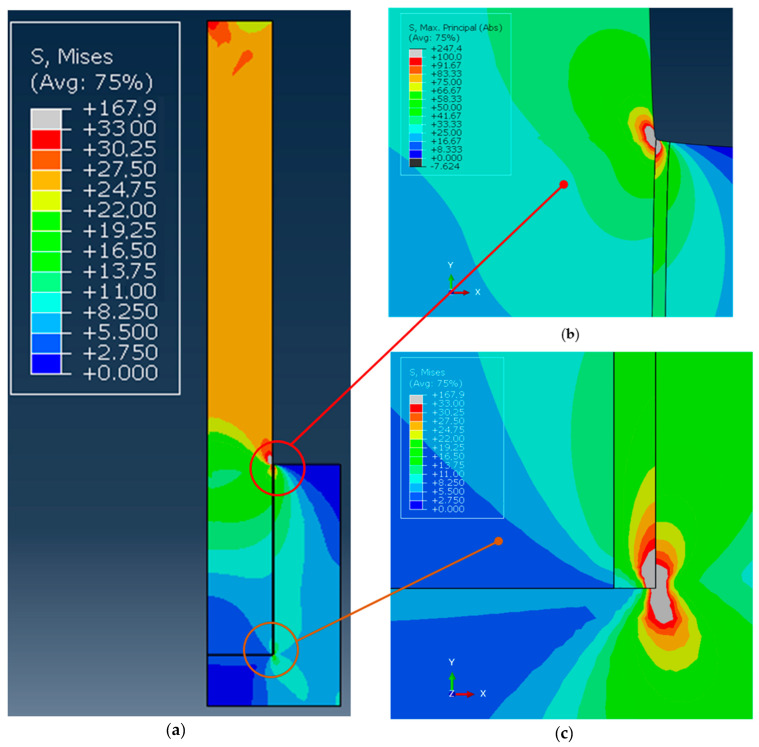
Von Mises stress maps of the CDHJ: (**a**) magnified absolute maximum principal stress in the adhesive upper corner (**b**) and magnified von Mises stress in the lower corner (**c**).

**Figure 13 polymers-16-03036-f013:**
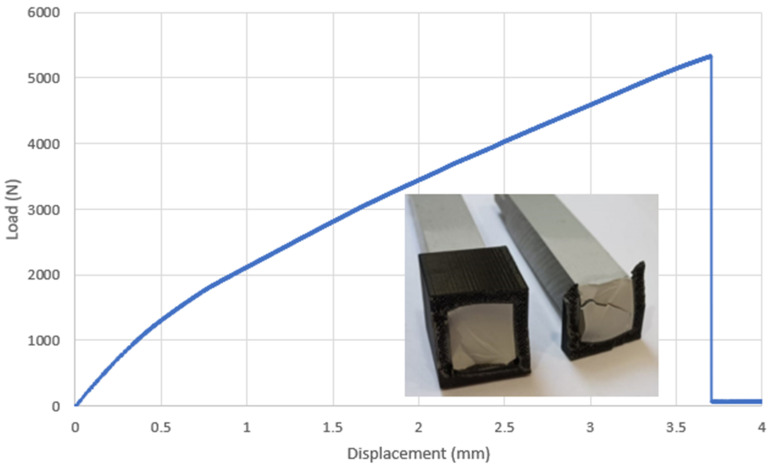
Load–displacement curve of the PDHJ with a fracture surface after joint failure.

**Figure 14 polymers-16-03036-f014:**
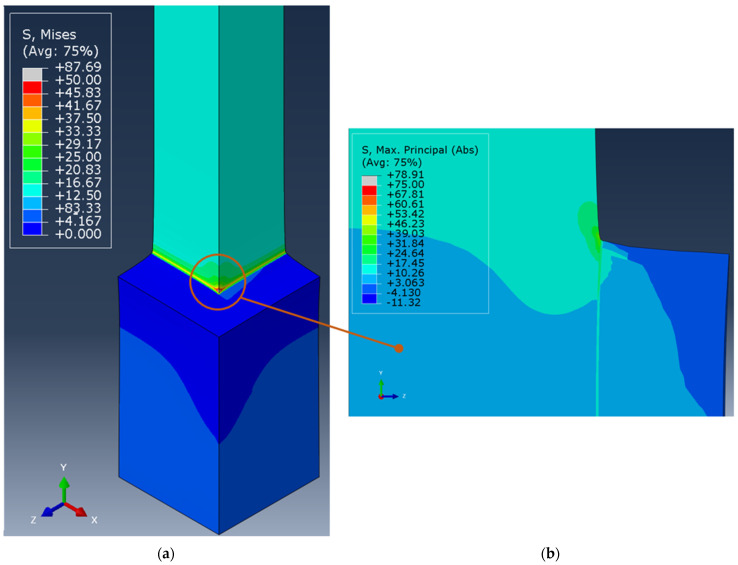
Von Mises stress contours of the PDHJ (**a**) and magnified absolute maximum principal stress in the adhesive corner (**b**).

**Table 1 polymers-16-03036-t001:** Average experimental values for peak force, shear stress, and stiffness of the pin–collar joints.

Exp. Values	Experimental Force	Av. Shear Stress	Stiffness	Displacement at Failure
Loctite 401	3595.7 N ± 15%	11.56 MPa ± 15%	4449 N/mm ± 5%	0.918 mm ± 18%
Loctite 480	1723.9 N ± 25%	5.22 MPa ± 25%	3649 N/mm ± 11%	0.466 mm ± 13%

**Table 2 polymers-16-03036-t002:** FE model results of the pin–collar joints.

	Applied Pressure	Max. Principal “*Structural*” Stress	Pin Max Vertical Displacement
Loctite 401	45.8 MPa	81.33 MPa	0.923 mm
Loctite 480	12.72 MPa	42.3 MPa	0.423 mm

**Table 3 polymers-16-03036-t003:** Comparison of the experimental and numerical values for the CDHJ.

	Peak Load	Displacement at Failure	Adhesive σ_critical_/σ_peak_	Adherend σ_critical_/σ_average_	Adherend σ_critical_/σ_peak_
Experimental tests	2129 N ± 6.8%	3.46 mm ± 4.1%	81.3/51.2 = 1.58	33/7.95 = 4.15	33/56.8 = 0.58
FE analysis	2130 N	3.1 mm			

## Data Availability

The original contributions presented in the study are included in the article/Appendix A, further inquiries can be directed to the corresponding author.

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
