# Peer review of "Mechanical Strength of Additive Manufactured and Standard Polymeric Components Joined Through Structural Adhesives"

_polymers, 2024, doi:10.3390/polym16213036_

Round 1
Reviewer 1 Report
Comments and Suggestions for Authors
In the article titled “Mechanical strength of additive manufactured and standard polymeric components through structural adhesives” presented by Spaggiari et al. authors have provided the analyses of mechanical properties of additive manufactured polymeric parts joined with standard plastic parts through structural adhesives. Overall the study is well described but there exist few issues which must be addressed before being published.
1. The abstract of the manuscript can be attractive for the readers if the authors mention the values of experimental values for peak force, shear stress and stiffness, finite element and Displacement at failure, in this way readers can easily understand the importance of this study. Moreover, abstract should be precise 250-300 words according to outcomes (suggestion).
2. Introduction section is more generally descriptive, there is no reference pointing out the literature outcomes (values) about peak force, shear stress and stiffness, finite element and Displacement at failure, problem statement is missing, why authors want to do this work, what was lacking in previous studies etc.
3. In the introduction section, out of 53 references only 6 references are (2021-2024) updated. I suggest the authors to provide the latest up-to-date survey which can better provide the comparison between the present study and previously published work (3D-Printed Conducting Polymers for Solid Oxide Fuel Cells. In3D Printed Conducting Polymers 2024 (pp. 179-195). CRC Press.).
4. There are many sentences in Page 3, “Experimental section” which must be revised as follow;
i. The pin collar specimens were designed as follows.
ii. Two different materials obtained from two different technologies were bonded. (which technologies, explain)
iii. The second adhesive is Loctite 480 [46] a toughened cyanoacrylic adhesive which provides, according……
iv. Moreover, the AM collars and were not sandblasted nor gritblasted before…….
I suggest the detailed English revision of the manuscript.
5. Section 2.1.2, “The numerical finite element models were designed using Abaqus (DS Simulia Established Products 2023) and exploiting the axisymmetric shape of the system.” What parameters were used for this calculation?
6. The model was intended to mainly tp provide a comparison, please explain the term “tp”
7. “The mesh dimension is comparable to the previous models,” please give the reference.
8. Figure 7 has the potential to be explained with details, please check its explanation.
9. Section 3.3, please index the equations and also mention them in the text with their little descriptions.
10. Conclusion should be revised, which should also be precise according to the outcomes.
Comments on the Quality of English LanguageEnglish of the manuscript is not strong, I suggest the English revision by the Native English speaker from same field or by the MDPI English editing service.
Author Response
Please refer to the attached file for the reviewers answers

Reviewer 2 Report
Comments and Suggestions for Authors
This study evaluates the mechanical properties of hybrid assemblies combining additive manufactured (AM) polymer parts with standard plastic parts using structural adhesives. The aim is to enhance the size and complexity of assemblies by leveraging AM for intricate joints, while using cost-effective extruded plastic parts for load-bearing components. The materials used include rigid and toughened acrylic adhesives, applied to ABS 3D-printed parts and PVC standard parts. The study first tests the adhesives using pin-collar joints, revealing that rigid adhesive performs better. In practical applications, the adhesive consistently transfers the load effectively, demonstrating that adhesive bonding can achieve similar strength to traditionally manufactured polymeric parts. This approach expands the potential of 3D printing for larger, hybrid structures.
Some revisions are necessary:
· Abstract: Proofread the abstract, as some punctuation is missing. Simplify the text to make it smoother and remove excessive technical details.
· Page 3: The sentence “Moreover, the AM collars and were not…” is unclear and needs revision to improve clarity.
· Figure 2: Since there's only one image in Figure 2, there's no need to refer to it as "Figure 2a." The text then mentions Figure 2b, suggesting something might be missing from Figure 2. Please check for completeness and proofread for any additional typos.
· Section 2.2.1: The statement “as reported by many researchers…” only cites one reference (number 50). This assumption should be supported by additional references or more explanation.
Moreover, only two replicates were manufactured. Is it enough to guarantee a good statistic, since that for other configurations 5 replicas were used?
· Figure 4: Some information is missing. Ensure completeness.
· Figure 7: The graph displays multiple blue and red curves representing Loctite 401 and 480, respectively. However, it is unclear what distinguishes the various curves. A clear legend describing the curves should be added to the graph or explained in the caption.
· Table in Figure 8c: The meaning of the numbers in this table is unclear. The results should be explained, with comments and references added where necessary.
- Paragraph 3.6: The first sentence is unclear and needs to be rewritten for clarity.
- Figure 14 caption: Review and correct the spacing issues.
- Supplementary Information: Proofread carefully. In Figure S1-S2, three replicas are presented. If these replicas are from the same type of sample, are the differences between them within the experimental or measurement error?
Overall, the authors should put more effort into refining the manuscript’s structure and thoroughly proofreading the text. Different fonts and text sizes are used inconsistently throughout the manuscript. Please standardize and make necessary adjustments.
Currently, the manuscript appears somewhat imprecise and requires extensive review.
Author Response
Please find all the answers to the reviewer comments in the attached file

Round 2
Reviewer 1 Report
Comments and Suggestions for Authors
I suggest authors have revised the manuscript well according to suggestions. Article can be published in current form.
Comments on the Quality of English LanguageEnglish of the manuscript is fine now.
Reviewer 2 Report
Comments and Suggestions for Authors
The issues were solved.